# Effect of the Integrated Addition of a Red Tara Pods (*Caesalpinia spinosa*) Extract and NaCl over the Neo-Formed Contaminants Content and Sensory Properties of Crackers

**DOI:** 10.3390/molecules27031020

**Published:** 2022-02-02

**Authors:** Franco Pedreschi, Joans Matus, Andrea Bunger, Romina Pedreschi, Nils Leander Huamán-Castilla, María Salomé Mariotti-Celis

**Affiliations:** 1Departamento de Ingeniería Química y Bioprocesos, Pontificia Universidad Católica de Chile, P.O. Box 306, Santiago 6904411, Chile; 2Departamento de Ciencia de los Alimentos y Tecnología Química, Facultad de Ciencias Químicas y Farmacéuticas, Universidad de Chile, P.O. Box 233, Santiago 8380000, Chile; joans.matus@ug.uchile.cl (J.M.); abunge@uchile.cl (A.B.); 3Escuela de Agronomía, Pontificia Universidad Católica de Valparaíso, Calle San Francisco s/n, La Palma, Casilla 4-D, Quillota 2263782, Chile; romina.pedreschi@pucv.cl; 4Escuela de Ingeniería Agroindustrial, Universidad Nacional de Moquegua, Prolongación Calle Ancash s/n, Moquegua 18001, Peru; nhuamanc@unam.edu.pe; 5Escuela de Nutrición y Dietética, Facultad de Medicina, Universidad Finis Terrae, Pedro de Valdivia 1509, Providencia, Santiago 7501015, Chile

**Keywords:** Maillard reaction, acrylamide, hydroxymethylfurfural, sensory properties, red Tara pods extract, NaCl, crackers

## Abstract

A 2k factorial design with three centrals points was considered to evaluate the effect of adding red Tara pods extract (Caesalpinia spinosa) (440–2560 µg/mL of dough water) and NaCl (0.3–1.7 g/100 g of flour) on the acrylamide (AA) and hydroxymethylfurfural (HMF) content and sensory attributes of crackers. Additionally, the best formulation, defined as that with the lowest AA and HMF content, was compared with a commercial formulation cracker. Red Tara pods extracts were obtained through conventional extraction using pure water (60 °C, 35 min). AA and HMF content were quantified by GC-MS and HPLC-DAD, respectively. The sensory evaluation was carried out using a descriptive analysis on a 10 cm non-structured linear scale. Red Tara pods extract significantly reduced (*p* < 0.05) the AA and HMF content, while NaCl only influenced the HMF formation. However, the sensory attributes did not significantly change (*p* > 0.05), excepting the violet-gray color and salty flavor, but at acceptable levels compared with the control sample. The higher the red Tara pods extract concentration (2560 µg/mL of dough water), the lower the neo-formed contaminants (NFCs) content of crackers (AA: 53 µg/kg and HMF: 1236 µg/kg) when salt level was below 3 g/100 g of flour. The action of the proanthocyanidins present in the extracts which trapped the carbonyl groups of sugars probably avoided the formation of both NFCs. Contrarily, NaCl addition (from 0.3 to 1.7 g/100 g of flour) significantly increased (*p* < 0.05) the HMF formation (from 1236 µg/kg to 4239 µg/kg, respectively), probably through the dehydration of carbohydrates during the Maillard reaction. When explored treatments were compared with a commercial formulated cracker, the highest mitigation effect (reductions of 40% and 32% AA and HMF, respectively) was reached with the addition of 2560 µg/mL of dough water of red Tara pods extract and 0.3 g/100 g of flour of NaCl. The addition of red Tara pods extracts integrated with the control of NaCl levels mitigated the NFCs in crackers, preserving their sensory properties. Future research should be focused on scaling this mitigation technology, considering a better chemical characterization of red Tara pods extracts as well as the validation of its use as functional food ingredient.

## 1. Introduction

Chile ranks the fourth in the international calorie consumption ranking, and the three main sources of calories consumed by Chileans are bakery products (bread, cookies, and breakfast cereals), oils and fats, and dairy products [1]. Interestingly, baked products have a large share in the Chilean food market, cookies and crackers being the most consumed products [1]. The development of attractive sensory attributes in these products is mainly due to the Maillard reaction (MR), in which the amino groups of proteins and the carbonyl groups present in carbohydrates interact at very high temperature (>120 °C) [2,3,4].

MR is one of the most important chemical reactions taking place during the frying, baking, roasting, and extrusion of foods [5]. Specifically, MR in protein–carbohydrate rich foods is important for the development of attractive color and flavor [6]. However, the major concern arising from high temperature processes applied to foods is the formation of toxic compounds such as acrylamide (AA) and hydroxymethylfurfural (HMF), called neo-formed contaminants (NFCs). NFCs may reveal harmful activities such as mutagenic, carcinogenic, and cytotoxic effects not only in vitro but also in in vivo trials [7,8]. AA is an amide produced as by-product of the MR in starchy foods processed at temperatures higher than 120 °C [9]. HMF is a furanic compound formed as an intermediate in MR when carbohydrates are heated in the presence of amino acids or proteins or, alternatively, by thermal dehydration of carbohydrates under acidic conditions [6].

Due to the high frequency of consumption of the high temperature processed starchy foods, different NFCs mitigation technologies have been proposed [5]. Most of these strategies are focused to disfavor the processing conditions under which MR is triggered [10]. One alternative is to decrease the processing temperature to reduce the activation energy of the reaction. Although this alternative is effective in food model systems, it is not scalable for real foods, in which the sensorial attributes developed during MR are crucial [11].

However, insights into the mechanisms of NFCs’ formation in thermally treated foods can also be altered by the addition of different compounds that affect the natural course of MR without decreasing the heating temperature [11].

For instance, the addition of antioxidants to food matrixes has been proposed as an effective alternative for NFCs mitigation. During the MR, polyphenols would interact with carbonyls and some key intermediaries of NFCs mainly by mechanisms of complexation through avoiding their formation [5,12,13].

In this sense, the replacement of synthetic food ingredients by safe natural alternatives is a trend worldwide. For the specific case of antioxidants, several new vegetable sources have been evaluated, obtaining promising results. This is the case of Tara (*Caesalpinia spinosa*), a South American leguminous tree whose pods contain high amounts of tannins (40% to 60% of the total content). Recently, Pedreschi et al. (2019) significantly mitigated the formation of some NFCs in Chilean bread by incorporating polyphenols extracted from Tara *(Caesalpinia spinosa*) pods [5]. These authors highlight the potential of using these polyphenolic extracts to reduce dietary NFCs exposure, which is of public health concern in Chile [11]. However, in many other cases, natural antioxidants have proved to be ineffective or even to promote the formation of AA, indicating that more research is required to define a recommended dose of the bioactive for each specific food matrix [14].

On the other hand, the mitigation effect of mono or divalent cations on the generation of NFCs is as well a pending issue [15]. Kolek et al. [16] proved the considerable inhibiting effects of sodium chloride (NaCl) on the formation of AA in asparagine–glucose model systems. In real foods, Pedreschi et al. [17] studied the effect of NaCl soaking after blanching in relation to the AA formation in potato slices after frying. They determined that soaking blanched potato slices in a 3 g/100 g NaCl solution for 5 min at 25 °C reduces AA formation by 11% in potato slices after frying. These authors postulated that blanching seems to make the NaCl diffusion in potato tissue easier, leading to a significant acrylamide reduction in the fried product. However, the salt consumption is a controversial issue, mainly because its excessive intake is related to several human metabolic diseases [1].

In this respect, different works performed in thermally treated food model systems found that the addition of di-valent cations such as Ca^2+^ increased HMF generation while decreasing AA formation [18,19]. NaCl addition presented a dual behavior depending on its concentration, suggesting that more research is necessary to elucidate the real effect of NaCl as a mitigation agent of NFCs [18].

To date, many effective ways to reduce the occurrence of NFCs in heat-treated and highly consumed starchy foods have been successfully established [12,17]. However, their effects on the sensory attributes of the final products have not been clearly elucidated [12,17]. Moreover, other relevant aspects of food design, such as maximum allowable levels for the addition of inhibitory agents, are not always considered, which would explain why these mitigation strategies have not yet been scaled up at the industrial level.

The objective of this research was to evaluate the effect of the integrated addition of an aqueous extract of red Tara pods (*Caesalpinia spinosa*) and NaCl on the occurrence of AA and HMF and the sensory attributes of crackers.

## 2. Materials and Methods

Red Tara pods harvested in March 2019 were kindly donated by the National Forest Corporation (CONAF, Santiago, Chile). Red Tara pods, received in a dried state, were manually cleaned, eliminating pods with fungal presence or damage by insects. Cleaned red Tara pods were ground using a mixer machine (model 4169/4297, Braun AG, Kronberg, Germany) until obtaining a fine powder, which was sieved through a 0.65 mm mesh sieve. Red Tara powder was stored in sealed polyethylene bags at ambient temperature and in absence of light until the solid–liquid extraction process was required.

### 2.1. Chemicals and Analytic Reagents

Analytical grade reagents, standards, and solvents were used in chemical analyses. Folin–Ciocalteu reagent, sodium carbonate, gallic acid, glucose, fructose, AA and HMF standards, dimethylaminocinnamaldehyde (DMAC; F.W. 175.23), solvents (acetone, methanol, acetonitrile, formic acid, hydrochloric acid, acetic acid, and ethanol), Carrez solution I, Carrez solution II, and sodium hydroxide were purchased from Sigma Aldrich (Steinheim, Germany).

### 2.2. Red Tara Pod Extract

Red Tara pods were ground and subjected to solid–liquid extraction according to a modified methodology of Pedreschi et al. [5]. The extraction was carried out with water (60 °C; solid/liquid ratio: 1/10) during 35 min with constant agitation. Then, the extract was centrifuged, and the supernatant was characterized in terms of antioxidant capacity and total polyphenol content. After that, the extract was concentrated (10 kPa) at 60 °C until being dried and stored in dark polyethylene bags (20 °C).

### 2.3. Preparation of Crackers

The crackers were prepared based on the formulation proposed by Aziha and Komathi [20], with some modifications (Table 1).

Briefly, different amounts of the dried red Tara pod extract (Table 2) were dissolved in the dough water. The water needed to make 100 g of dough was determined considering that the flour contained 14% moisture. In addition, different levels of NaCl addition per 100 g of flour (Table 2) were set within the limits required by the Chilean food labeling law [21], to ensure that the final product was not categorized as “High in Sodium”.

The flour, the reconstituted red Tara pods extract, the salt, and the butter were added to a mixer machine (Pro Line^®^ Series 7 Quart Bowl-Lift Stand Mixer, KitchenAid, Troy, OH, USA), and yeast previously dissolved in warm water (37 °C) was added. The mixing process was carried out until obtaining a homogenous dough. The dough was then left to ferment covered with plastic film at about 25 °C for 3 h to allow the flavors and textural characteristics typical of the final product to develop. Subsequently, the resulting dough was laminated (thickness of about 3 mm) and crackers were shaped into cylinders (30 cm in diameter and 3 mm wide). The pieces were baked in a forced convection oven at 180 °C for 25 min. After baking, crackers were cooled and storage in low density polyethylene sealed bags. The cookies for chemical analysis were kept frozen at approximately −20 °C, and those destined for sensory evaluation were stored at room temperature (20 °C) for no more than 1 day until the time of evaluation.

### 2.4. Chemical Analysis

#### 2.4.1. Total Polyphenol Content (TPC)

The TPC of red Tara pods extract was determined by the spectrophotometric Folin−Ciocalteu assay (Spectrometer UV 1240, Shimadzu, Kioto, Japan) and expressed as g of gallic acid equivalent (GAE) per g of dry Tara pods [22].

#### 2.4.2. Antioxidant Capacity (AOC)

The AOC of red Tara pods extract was determined by spectrophotometry (Spectrometer UV 1240, Shimadzu, Kioto, Japan) according to the free radical 2,2-diphenyl- 1-picrylhydrazyl method (DPPH) and expressed as the efficient concentration of extract (EC_50_: mg/mL), which is the concentration necessary to inhibit 50% of radical absorption of DPPH [23].

#### 2.4.3. Determination of the Acrylamide (AA) Content

The AA content of crackers was determined by gas chromatography coupled with mass spectrometry (GC- MS) according to the methodology described by Barrios et al. [11].

#### 2.4.4. Determination of the 5-Hydroxymethylfurrfural (HMF) Content

The HMF content of crackers was measured by high-performance liquid chromatography coupled with diode array detector (HPLC-DAD) according to the methodology described by Pedreschi et al. [5]. Sample preparation consisted of the aqueous extraction of 1 g of sample in 20 mL of distilled water. Removal of interferents was performed with 1.5 mL of Carrez reagent I (K4[Fe (CN)6] ×3H_2_O) and 1.5 mL of Carrez reagent II (ZnSO_4_ × 7H_2_O) by shaking at each addition and centrifuging for 15 min. Then, the supernatant was obtained and passed through a 0.22 µm nylon filter into 2 mL vials for subsequent injection. The HPLC-DAD conditions were: (i) mobile phase: 1% acetic acid/acetonitrile (95/5% ratio), (ii) mobile phase flow rate: 1 mL/min, (iii) detector wavelength: 284 nm, (iii) injection volume: 20 µL, and (iv) column: AcclaimTM 120 C18 5 µm 120 Å 4.6 × 150 mm.

### 2.5. Sensory Evaluation

Training of the sensory panel was performed in the Sensory Evaluation Laboratory of the Universidad de Chile according to the standard ISO 8586:2012. 12 [24]. Assessors with experience in sensory testing were selected: 7 females and 5 males, all students of food engineering with ages between 23 and 28 years old. Their training process consisted in one initial session to generate descriptors and five 1 h sessions for assessing samples with different Tara extract concentrations, salt concentrations, and baking times.

A descriptive analysis on a 10 cm non-structured linear scale was used, considering the following descriptors directly related to the addition of Tara extract, salt concentration, and baking time (anchors between parenthesis): toasted color (light–dark), grey violet color (non-strong), toasted aroma (non-strong), crispness (not crisp–very crisp), hardness (soft–hard), toasted flavor (non-strong) and saltiness (non-strong). The descriptor “grey violet color” was developed by Pedreschi et al. [5] during development of a descriptive analysis of bread added with a similar Tara extract.

The performance of the sensory panel and of each panelist was validated following the guidelines of the standard ISO 11132:2012 [25]. For the key descriptors violet-grey color and salt were assessed by testing three significantly different samples using the following Tara extract concentrations and salt concentrations, respectively: (i) 500 µg/mL and 0 mg/100 g flour; (ii) 1500 µg/mL and 2 g/100 g flour; (iii) 3000 µg/mL and 1 g/100 g flour. Samples were evaluated in triplicate in separate sessions.

With the trained and validated panel, the 5 treatments of the experimental design were assessed by descriptive analysis. Samples were presented in random order, evaluating 3 to 4 samples per session. Filtered water was provided to cleanse the palate between samples.

### 2.6. Experimental Design and Statistical Analysis

A 2k factorial with three centrals points design was considered to assess the effect of red Tara pod extract and salt (NaCl) addition on the acrylamide content, HMF content, and descriptive profile. Results are presented as mean and coefficient of variation. The analysis of variance (ANOVA) and Tukey test were applied to the response variables (*p* < 0.05). After, the best condition was compared with a commercial formulated product. A paired Student’s t test was performed to establish differences between both conditions on the different response variables (*p* < 0.05). Statgraphics Plus statistical program for Windows 4.0 (Statpoint Technologies, Inc., Warrenton, VA, USA) was used to analyze the data.

## 3. Results and Discussion

### 3.1. Characterization of the Red Tara Pods Extract

The obtained extract was characterized in terms of yield, TPC, and AOC (expressed as EC_50_). Yield, defined as the ratio between the weight of Tara extract obtained and the weight of milled red Tara pods used for the extraction, was 43%. It agreed with values previously reported for the same raw material where it was described that milled pods concentrated tannin content between 40–60% (*w/w*) [26].

Regarding chemical characterization, the obtained extract presented a TPC of 663.1 mg GAE/g and an EC_50_ of 8.01 µg/mL. Previous studies carried out with the same raw material but under different extraction conditions reported different values for both antioxidant properties. For instance, an aqueous macerate of powdered pods presented a TPC of 149 mg GAE/g and an EC_50_ of 10.1 µg/mL [27], while other hydroalcoholic extracts of Tara presented EC_50_ values of 4.52 μg/mL [28] and 4.97 μg/mL [29]. However, Bravo [30] and Lopez et al. [31] who used the same extraction method applied in our work obtained TPC of 473.4 mg GAE/g and 563.7 mg GAE/g, respectively.

Based on the obtained results, we can affirm that the TPC of the red Tara pods extract is higher than reported in the literature, while its antioxidant activity, although somewhat elevated, is still within the expected range. Thus, the extract obtained is considered suitable for its use as potential mitigation agent of NFCs.

### 3.2. Effect of the Integrated Addition of the Red Tara Pods Extract and NaCl on the Acrylamide and HMF Content of Crackers

This study evaluated the effect of adding different amounts of an aqueous extract of red Tara pods and NaCl together on the AA and HMF content of crackers. Red Tara pods extract and NaCl addition significantly affected (*p* < 0.05) the HMF occurrence under all evaluated conditions, while the AA formation was only significantly influenced (*p* < 0.05) by changes in the polyphenolic extract concentration (Table 3).

The higher the red Tara pods extract concentration, the lower the AA final content of crackers (Table 3). Interestingly, for the HMF mitigation the red Tara pods extract concentration was influenced by the NaCl level of addition. At the lowest NaCl addition level (0.3 g/100 g), an increase in red Tara pod addition significantly reduced the HMF content of cracker (Table 3). Contrarily, when the highest level of NaCl addition was added (1.7 g/100 g), an increase in the red Tara pod extract concentration did not significantly reduce the HMF content of cracker (Table 3).

The association between the antioxidant activity of polyphenols and their ability to mitigate the AA and HMF formation is still a controversial subject. The various effects of polyphenols on NFCs formation and/or inhibition are related to their structure, concentration, antioxidant capacity, and reaction conditions [32,33,34,35,36]. The terminal functional groups of the side chain, hydroxyls, and aldehydes of polyphenols would be key in their ability to disrupt or enhance certain steps in the AA and HMF formation pathways [36,37,38].

Some polyphenols such as proanthocyanidins can significantly and dose-dependently inhibit the formation of AA and HMF, probably through trapping of carbonyl compounds, preventing against lipid oxidation or reacting with the electron-deprived vinyl double bond present in the AA molecule, removing it [33,35,36,39].

Contrarily, other ones such as gallic and chlorogenic acids provide carbonyl groups that promote both the HMF formation and the conversion from 3-aminopropionamide (3-APA) to AA [38,40].

In this sense, red Tara pods are characterized by their high content of both hydrolysable and condensed tannins. The hydrolysable tannins are readily hydrolyzed in phenolic acids such as gallic and chlorogenic acids, while condensed tannins or proanthocyanidins are polyflavonoids in nature, consisting of chains of flavan-3-ol units. Therefore, studies on the mechanism of action of red Tara pod polyphenols as potential inhibitors of NFCs are ongoing and should be continued [41,42].

Interestingly, at the evaluated experimental region, NaCl addition did not significantly decrease (*p* > 0.05) the AA content of crackers (Table 3). NaCl had shown an AA decreasing potential in food model systems. Levine et al. [19] observed in a bulk model system that AA concentration decreased with increasing NaCl concentration. Similarly, Gökmen and Şenyuva [18] found in a liquid model system consisting of 10 µmol/L asparagine and glucose that the final AA concentration was higher in the control, decreasing at NaCl concentrations from 0.5 to 5 µmol/L and decreasing significantly for concentrations from 5 to 20 µmol/L. However, bakery products represent a considerably more complex matrix, with numerous factors affecting AA formation. In this sense, it was evidenced that the optimal NaCl concentration to minimize AA levels in bread was between 1% and 2%, which is usually used in industrial bread production [43]. Along with this, the reduction in AA in food model systems using higher levels of NaCl is not transferable to Chilean bakery products, mainly, due to technological aspects such as the inhibition of the fermenting yeast and some changes in the sensory attributes of the final product and due to the recommendations of the Food Labeling Chilean Law regarding the maximum sodium limits established for processed foods [21].

Contrary to the evidenced effect of NaCl addition on the AA mitigation in crackers, it significantly increased the generation of HMF (Table 3). HMF occurrence in thermally treated starchy foods is more clearly related to NaCl concentration. Monovalent cations favor the dehydration of HMF’s key intermediates and hexoses, leading to an increase in its formation [18,44]. Both fructofuranosyl cation—an HMF key intermediate originated from sucrose—and glucose can generate HMF through the removal of 3 and 2 moles of water, respectively. When the concentration of certain cations is increased, the reaction is mainly directed towards the dehydration of glucose, giving rise to HMF as one of the characteristic end products [45].

When explored treatments were compared with the commercial formulated cracker, the highest mitigation effect (reductions of 40% and 32% of AA and HMF, respectively) was reached with the addition of 2560 µg/mL of dough water and 0.3 g/100 g of flour of red Tara pods extract and NaCl, respectively (Figure 1). Under these addition levels, it was possible to elaborate crackers reduced in NFCs which did not overcome the sodium limit establish by the Chilean food regulation [21].

### 3.3. Effect of the Addition of the Red Tara Pods Extract and NaCl on the Descriptive Profile of Crackers

The application of bioactive ingredients obtained from by-products in the rational design of new healthy foods represents a key strategy for the future. In this sense, the addition of red Tara pods extracts integrated with the control of the NaCl levels successfully mitigated the NFCs in crackers. However, although the nutritional value and safety are main issues to optimize in the development of functional foods, the real scalability of an innovative food process also depends on the sensorial attributes of the final product. Therefore, the descriptive profile of crackers added with different amounts of red Tara pods extract and NaCl was evaluated (Table 4).

The main sensory differences between samples corresponded to the descriptors that depend on the factors that are variable in the experimental design: “grey violet color” and “saltiness” (Table 4), with direct relation to the concentration of red Tara pods extract and NaCl, respectively. The external toasted color, toasted aroma, crispness, hardness, and toasted flavor did not present differences between samples, showing that the addition of a red Tara pods extract did not affect attributes such as texture or flavor of the crackers. In this sense when the best treatment (lowest AA and HMF levels) was compared with the commercial formulated crackers, the same behavior was observed, with “grey violet color” and “saltiness” being the descriptors with significant differences (Figure 2).

Contrarily, the addition of an extract of Rocha pear peels to wheat and rye breads provoked slight differences in the sensory quality of both products. The polyphenolic extract decreased the acrylamide content of both bread types (27.3% and 19.2%, respectively). However, the breads with extracts addition presented lower scores in the hedonic evaluations compared with the control sample [46].

These conflicting results could be attributed to the differences in the phenolic profile of the red Tara pods and Rocha pear peels extracts. Depending on the polyphenol type, specific bonds can be formed with the aromatic compounds generated during Maillard reaction, which obviously differentially influence not only the flavor but also the aroma of the baked starchy products [40]. In this regard, the addition of different pure polyphenols to a bread model system adversely affected its volatile profile. The phenolic compounds reduced the acrylamide content (16.2–95.2%), but at the highest addition level (2.0%), caffeic acid most significantly suppressed Maillard-type volatiles (75.9%), followed by gallic acid (74.3%), ferulic acid (65.6%), (+)-catechin (62.4%), and quercetin (59.3%) [40]. Therefore, the future development of these acrylamide mitigating agents, must consider the fully chemical characterization of the polyphenolic extract to potentially predict interactions with aromatic compounds naturally present in the food matrix. Moreover, the form selected for adding the extract, such as aqueous or dry, should be also considered, as well as the type of oven used for baking, considering that both polyphenols and Maillard compounds are thermolabile.

## 4. Conclusions

The use of high concentrations of polyphenol-rich Tara extracts (2560 µg/mL) combined with low levels of NaCl (0.3 g/100 g) allowed a 2- and 5-fold reduction in the AA and HMF contents of crackers compared with a commercial formulated product. Moreover, the antagonistic effect of both study factors (polyphenols and NaCl) did not affect the sensory properties of the original food.

The higher the red Tara pods extract concentration, the lower the AA and HMF occurrence in crackers, which was probably due to the action of tannins contained in the extracts that trapped the Maillard intermediates of both NFCs.

Interestingly, at the NaCl added levels, the AA content of crackers was not significantly reduced. It can be attributed to an inhibition effect caused by the salt on the fermenting yeast that increased the available reducing sugars for the Maillard reaction. Moreover, the NaCl addition significantly increased the HMF under all evaluated conditions, mainly due to the dehydration of carbohydrates, which favor its generation.

Therefore, the addition of red Tara pod extracts integrated with the control of NaCl content is a feasible alternative to elaborate tasty crackers reduced in NFCs that respect the sodium limit establish by the Chilean food regulation.

The scaling of this NFCs mitigation technology seems to be a good alternative not only for improving the safety of highly consumed starchy foods but also for the commercialization of polyphenol-rich foods with sensory qualities equivalent to those of conventional products. In this sense, a better chemical characterization of the red Tara pod extract is highly desirable to improve the understanding of the mechanisms associated with its mitigation effect, as well as its potential functional applications.

## Figures and Tables

**Figure 1 molecules-27-01020-f001:**
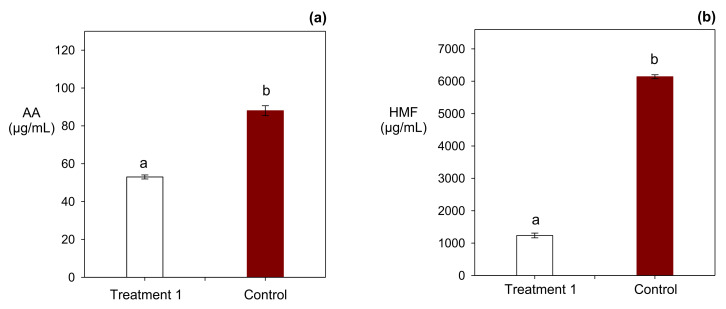
Effect of the red Tara pod and NaCl addition on the AA and HMF content of crackers. Different low case letters indicate statistically significant differences (*p* < 0.05) between the control sample and the treatment 1 (**a**) AA content of control and treatment 1 (**b**) HMF content of control and treatment 1.

**Figure 2 molecules-27-01020-f002:**
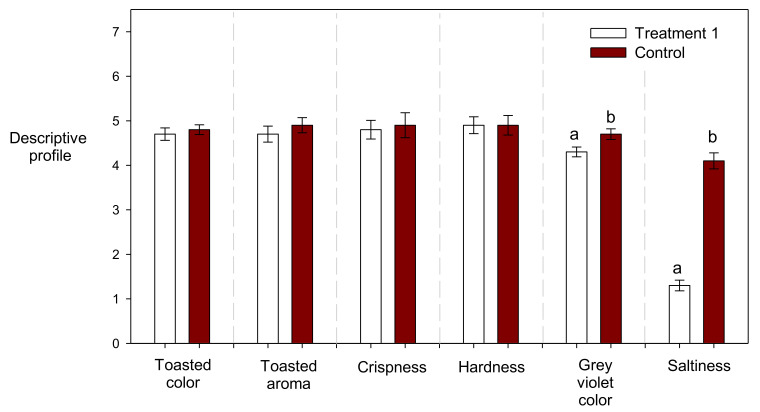
Descriptive profile of a commercial formulated cracker (control) and reduced AA and HMF crackers. Different low case letters indicate statistically significant differences (*p* < 0.05) between the control sample and the treatment 1.

**Table 1 molecules-27-01020-t001:** Formulation of crackers.

Ingredient	Quantity (g)
Flour	100
Salt	0–2 *
Lard	7
Yeast	3.5
Water	45

* Salt corresponds to one variable in the experimental design, and it varies in the presented range according to Table 2.

**Table 2 molecules-27-01020-t002:** Evaluated operational conditions.

Treatments	Red Tara Pod Extract (µg/mL) *	NaCl (g/100 g) **
1	2560	0.3
2	440	0.3
3	2560	1.7
4	440	1.7
5	1500	1
6	1500	1
7	1500	1

* mL of water used in dough elaboration. ** g of wheat flour.

**Table 3 molecules-27-01020-t003:** Acrylamide and HMF content of crackers with different addition levels of red Tara pods extract and NaCl.

Treatment	Polyphenolic Extract (µg/mL) *	NaCl (g/100 g) **	HMF (µg/kg) ***		Acrylamide (µg/kg) ***	
			Mean	CV	Mean	CV
1	2560	0.3	1236 ^a^	0.06	53.00 ^a^	0.02
2	440	0.3	2544 ^b^	0.05	72.33 ^b^	0.02
3	2560	1.7	4239 ^d^	0.01	50.33 ^a^	0.08
4	440	1.7	4184 ^d^	0.01	71.00 ^b^	0.05
5	1500	1	3902 ^c^	0.02	69.67 ^b^	0.02
6	1500	1	3928 ^c^	0.02	66.67 ^b^	0.02
7	1500	1	3920 ^c^	0.06	68.00 ^b^	0.01

The results are expressed as the mean and CV (coefficient of variation). Different letters indicate statistically significant differences (*p* < 0.05). * mL of water used in dough elaboration. ** g of wheat flour. *** kg of cracker.

**Table 4 molecules-27-01020-t004:** Descriptive profile of crackers added with different amounts of red Tara pods extract and NaCl.

Treatments	Polyphenolic Extract (µg/mL) *	NaCl (g/100 g) **	Toasted Color	Grey Violet Color	Toasted Aroma	Crispness	Hardness	Toasted Flavor	Saltiness
1	2560	0.3	4.7	4.3 ^b^	4.7	4.8	4.9	4.9	1.3 ^a^
2	440	0.3	4.8	2.9 ^a^	4.7	4.8	4.8	4.9	1.4 ^a^
3	2560	1.7	4.9	4.8 ^c^	4.9	5.2	5.1	4.9	5.0 ^c^
4	440	1.7	4.9	2.9 ^a^	4.9	5.0	5.0	5.0	4.9 ^c^
5	1500	1	4.9	4.3 ^b^	4.7	4.9	4.8	4.9	4.2 ^b^
6	1500	1	4.9	4.2 ^b^	4.7	4.9	4.8	4.9	4.2 ^b^
7	1500	1	4.9	4.2 ^b^	4.7	4.8	4.9	4.9	4.1 ^b^

Results are expressed as the mean (*n* = 9). Different low case letters in the same column indicate statistically significant differences (*p* < 0.05). * mL of water used in dough elaboration. ** g of wheat flour.

## Data Availability

Not applicable.

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
