# Peer review of "Effect of the Integrated Addition of a Red Tara Pods (Caesalpinia spinosa) Extract and NaCl over the Neo-Formed Contaminants Content and Sensory Properties of Crackers"

_molecules, 2022, doi:10.3390/molecules27031020_

Round 1

Reviewer 1 Report

This work deals with the effects of adding an red Tara pod extracts and NaCl on the neo-formed contaminants, such as acrylamide and hydroxymethylfurfural contents, as well as sensory attributes of crackers. Experiments are well done and designed.  

The authors put a lot of effort into the monitoring of particular content, but there is no significant scientific contribution, as well as appropriate discussion of obtained results and effects. I recommended the addition of possible chemical reactions of processes therein.

However, the total phenolic content and antioxidant activity of Tara pods are monitored before thermal treatment, it would be interesting to see results after thermal treatment (in crackers), due to a lot of phenolic compounds are thermo-degradable, and it could influence neo-formed contaminants content.

I recommend publication of “Effect of the integrated addition of a red Tara pods (Caesalpinia 2 spinosa) extract and NaCl over the neo-formed contaminants content and sensory properties of crackers”, after the deeper discussion of obtained effects, as well as inclusion in the manuscript of possible chemical reactions of process found.

Author Response

Response: The discussion was improved according to your suggestions including a deeper discussion of obtained effects and possible chemical reactions of process found, please see lines (26-28 and 268-286).

In this sense, to determine the effect of baking on the antioxidant capacity and total polyphenol content of crackers added with red Tara pod extracts could contribute to validate that AA and HMF mitigation effect is caused by their addition.

However, we think that this is another study, in which it is also necessary to include the determination of specific polyphenols additionally to the total polyphenol content and antioxidant capacity.

In previous studies we found that these two antioxidant properties are not specific and not always present a good correlation with the change in the content of specific polyphenols. Moreover, the HMF content was directly correlated with antioxidant capacity (DPPH) and Total polyphenol content (Mariotti et al., 2017;2018; Huaman et al., 2019; 2020).

Additionally, due to sanitary situation in Chile it is not possible to carry out more experimental work until the National vaccination plan has been completed.

Reviewer 2 Report

The study addressed a subject of great interest in the area of food toxicology regarding the attenuation of the generation of NFCs through the addition of an extract rich in antioxidants and NaCl, and its implication in sensory properties.

Although it is not entirely novelty (the use of Tara extract - Caesalpinia spinosa, due to its antioxidant properties), the results obtained in the study with crackers provide an advance in current knowledge. It should be noted that the group has published several works on this topic. When it comes to the addition of plant extract, I emphasize that the plant material should be correctly typed (in italics). Therefore, the entire text should be revised and corrected.

The introduction is well written, exposing the problem in a clear and objective way. The use of an experimental design enriches the study, providing a solid statistical basis for the experiments, but there are some questions/suggestions to be made:

  • Line 207: 2k factorial design or 22 factorial design. The authors cited “...with three central points” but in fact, only one test was carried out under experimental conditions (correct???? – test 5) and not 3 central points (which would total 7 experiments). Please, correct it. Repeating the center point was to obtain an estimate of the experimental error. If more experiments were performed at the midpoint, complete Table 3, please.
  • Line 209: response variables were acrylamide content and HMF content (descriptive profile should be removed)
  • Line 251-266: the text should be revised because “The higher the red Tara pods extract concentration, the lower the AA ande HMF final contents of crackers IT IS NOT TRUE FOR HMF (ONLY AA) SEE TREATMENT #3, PLEASE. Lines 327-331 are correct!!! Please, make the necessary corrections.

In fact, from line 251 onwards, the text must be rewritten, explaining the results obtained in the experimental design. Here, it is worth exploring the planning results better (was there an interaction between the variables???? In other words, does the effect of one variable depend on the level of the other? The existence of interaction makes the isolated analysis of the effect of a single factor incorrect.

Since there were only 2 variables in the study (polyphenolic extract and NaCl) Why didn't the authors choose to carry out a central composite design, with the axial points and adjust a mathematical model to the experimental data? With this it would be possible to obtain response surfaces and the derivation of the mathematical model would estimate the optimal point to be proved experimentally. There would be few additional trials.

  • Line 376-377, please, correct the text (not to HMF, see treatment #3)
  • Line 294: I didn't understand where these values came from (AA:40% and HMF:32%)
  • Line 173-176: According to the authors, “Determination of HMF was measured by HPLC-DAD according to the methodology described by Pedreschi et al. [5]”, but Pedreschi used the method of Toker et al. 2013 developed to molasses. And Toker et al. 2013 used the method of Zappala et al 2005 to honey samples. It is recommended that the authors provide an accurate description of the method of sample preparation and chromatographic analysis for HMF quantification.

Minor corrections:

- Line 136: give spacing between the numerical value and the unit (standardize in the entire manuscript)100 g not 100g.

- Line 221:  3. Results and discussion (not number 2) and in line 371 Conclusions are number 4 (not 5).

- There are 2 separate tables identified as Table 3 (pages 7 and 9)

- In page 6 lines 234 and 236, correct in the text: Avilés et al. (2010); Bravo (2010) e Lopez et al. (2011)  and include in the references.

- Line 225: AOX???? AOC???? Please correct it.

- correct micrograms µg not ug (in all text)

Author Response

Comments and Suggestions for Authors

The study addressed a subject of great interest in food toxicology regarding the attenuation of the generation of NFCs through the addition of an extract rich in antioxidants and NaCl, and its implication in sensory properties.

Although it is not entirely novelty (the use of Tara extract - Caesalpinia spinosa, due to its antioxidant properties), the results obtained in the study with crackers provide an advance in current knowledge. It should be noted that the group has published several works on this topic. When it comes to the addition of plant extract, I emphasize that the plant material should be correctly typed (in italics). Therefore, the entire text should be revised and corrected.

Response: Plant material (Caesalpinia spinosa) was correctly typed in italics in the entire document.

The introduction is well written, exposing the problem in a clear and objective way. The use of an experimental design enriches the study, providing a solid statistical basis for the experiments, but there are some questions/suggestions to be made:

  • Line 207: 2k factorial design or 2factorial design. The authors cited “...with three central points” but in fact, only one test was carried out under experimental conditions (correct???? – test 5) and not 3 central points (which would total 7 experiments). Please, correct it.

Response: As you mentioned, we performed a 2k factorial design with 3 central points, consequently 7 treatments were performed. This was corrected in tables 2, 3 and 4.

  • Repeating the center point was to obtain an estimate of the experimental error. If more experiments were performed at the midpoint, complete Table 3, please.

Response:  Table 3 was corrected with all the experiments performed. Additionally, we included the determinations of AA and HMF as well as sensory evaluation of the control sample.

On the other hand, we should mention that although the use of the central points allows estimating the experimental error, their use should also be considered to detect the curvature between the levels proposed for this study (red Tara pod extract and NaCl concentrations). This will allow establishing whether the central point is higher or lower on average with respect to the other levels of the factorial design, and consequently a new research can be established in the future to optimize the doses of red Tara pod extract and NaCl.

  • Line 209: response variables were acrylamide content and HMF content (descriptive profile should be removed)

Response: Response variables evaluated at the 5 different conditions were AA, HMF and descriptive profile (Table 3 and Table 4). We changed the redaction of the document to improve its understanding (Lines 215-223).

  • Line 251-266: the text should be revised because “The higher the red Tara pods extract concentration, the lower the AA ande HMF final contents of crackers IT IS NOT TRUE FOR HMF (ONLY AA) SEE TREATMENT #3, PLEASE. Lines 327-331 are correct!!! Please, make the necessary corrections.

Response: Regarding to your comment, in fact when you observe only the experiment number 3, the cracker with the highest Tara pod extract concentration did not present the lowest HMF content.

However, if you observe the table 3, “at the lowest NaCl level (0.3 g/100 g), an increase of Red Tara pod addition (from 440 µg/ml to 2560 µg/ml), significantly reduced the HMF content of cracker (from 4184 to 1236 µg/Kg). Contrary when the highest level of NaCl addition was added (1.7 g/100 g), an increase in the Tara pod extract addition (from 440 µg/ml to 2560 µg/ml) did not significantly reduce the HMF content of cracker”.

This explanation was included in the corrected version of the paper to clarify the discussion of the results (Lines 261-267).

  • In fact, from line 251 onwards, the text must be rewritten, explaining the results obtained in the experimental design. Here, it is worth exploring the planning results better (was there an interaction between the variables???? In other words, does the effect of one variable depend on the level of the other? The existence of interaction makes the isolated analysis of the effect of a single factor incorrect.

Response: It was rewritten (Lines 261-267). Regarding to your question, at the highest level NaCl influence the Red Tara pod effect on the HMF mitigation.

  • Since there were only 2 variables in the study (polyphenolic extract and NaCl) Why didn't the authors choose to carry out a central composite design, with the axial points and adjust a mathematical model to the experimental data? With this it would be possible to obtain response surfaces and the derivation of the mathematical model would estimate the optimal point to be proved experimentally. There would be few additional trials.

Response: As you mentioned it would be very useful to carry out an optimization for defining the optimal point and then validate experimentally the mathematical model. However, this is an exploratory study, which evaluated the integrated effect of NaCl and red Tara pod extract addition not only on the AA and HMF contents in crackers but also on the sensory properties of this food. Obtained results in this research allowed to define the experimental region for developing an optimization in a future study which considers a deeper sensorial evaluation using more specific quality tests. Although it is possible to optimize only the mitigation of AA and HMF, we consider that this knowledge without sensory response is not transferable for real foods. Additionally, considering the sanitary scenery in our country it is not possible to develop consumer quality evaluations and more experiments at laboratory.

  • Line 376-377, please, correct the text (not to HMF, see treatment #3)

Response: It is possible to reduce both the AA and HMF contents of crackers preserving their sensorial properties, through the combined effect of (i) controlling the NaCl level and (ii) adding a red Tara pod extract of the cracker formulation. It can be observed for the treatment 1 of table 3 and 4.

  • Line 294: I didn't understand where these values came from (AA:40% and HMF:32%)

Response: These values represent the AA and HMF percent of reductions reached in treatment 1 compared with a control cracker (0 µg/ml red Tara pods extracts and 1 g/100g of NaCl). Both the NFCs and sensory profile of the control sample were included in Figure 1 and 2 and clarified in the document (Lines 323-326).

  • Line 173-176: According to the authors, “Determination of HMF was measured by HPLC-DAD according to the methodology described by Pedreschi et al. [5]”, but Pedreschi used the method of Toker et al. 2013 developed to molasses. And Toker et al. 2013 used the method of Zappala et al 2005 to honey samples. It is recommended that the authors provide an accurate description of the method of sample preparation and chromatographic analysis for HMF quantification.

Response: The HMF quantification method was described as follow (Lines 177-184).

“HMF was measured by HPLC-DAD according to the methodology described by Pedreschi et al. [5]. Sample preparation consisted in the aqueous extraction of 1 g of sample in 20 mL of distilled water. Removal of interferents was performed with 1.5 mL of Carrez reagent I (K4[Fe (CN)6]x3H2O) and 1.5 mL of Carrez reagent II (ZnSO4 x 7H2O) by shaking at each addition and centrifuging for 15 min. Then, the supernatant was obtained and passed through a 0.22 µm nylon filter into 2 mL vials for subsequent injection. The HPLC-DAD conditions were: (i) mobile phase: 1%acetic acid /acetonitrile (95/5% ratio), (ii) mobile phase flow rate: 1 mL/min, (iii) detector wavelength: 284 nm, (iii) injection volume: 20 µL and (iv) column: AcclaimTM 120 C18 5 µm 120 Å   4.6x 150 mm”

Minor corrections:

  • Line 136: give spacing between the numerical value and the unit (standardize in the entire manuscript)100 g not 100g.

Response: It was done, please see line 137.

  • Line 221:  3. Results and discussion (not number 2) and in line 371 Conclusions are number 4 (not 5).

Response: It was done (Lines: 231 and 394).

  • There are 2 separate tables identified as Table 3 (pages 7 and 9).

Response: It was corrected, please see lines 320 and 351 of the corrected document.

  • In page 6 lines 234 and 236, correct in the text: Avilés et al. (2010); Bravo (2010) e Lopez et al. (2011) (Lines 245 and 246) and included in the references.

  • Line 225: AOX???? AOC???? Please correct it.

Response: It is AOC (Line 235).

  • correct micrograms µg not ug (in all text)

Response: It was corrected in the entire document

Reviewer 3 Report

The manuscript is on an interesting subject, however, following points should be carefully addressed. Abstract: Include numerical values of results within abstract. Mention the methods used for study. What variations come after addition of extract and NaCl quantify the change in abstract. Also mention the future application at the end of abstract. Methodology: Why extraction was carried out for only 35 minutes for pods??? 35 mins by conventional extraction is too short to exhaust the material for valuable metabolites. Mentioned atleast references at first appearance for the methods of glucose, fructose, antioxidant activity and Total phenolic contents estimation. Authors should briefly describe tannin contents in tara pods. Results: Line 270-315, reference style is not as mentioned in guidelines to authors and the variations Line 332, ins?? Table 3 caption is mentioned twice, moreover why table 3 does not indicate mean and SDs?? The manuscript lack in sufficient statistical analysis of the data, specifically tables of the manuscript. Conclusions: The text is repetition of results. authors should mentioned concrete outcomes, future applicability and significance of the study. Discussion needs more evidence support from the literature.

Author Response

Reviewer 3:

Dear reviewer 3 thanks in advance for all your comments and suggestions. We included all of them in the manuscript significantly improving its quality and understanding.

All the changes were highlighted in red and the answer to your questions are in the following paragraphs:

The manuscript is on an interesting subject, however, following points should be carefully addressed.

  • Abstract: Include numerical values of results within abstract. Mention the methods used for study. What variations come after addition of extract and NaCl quantify the change in abstract. Also mention the future application at the end of abstract.

Response: All your comments were considered, and the abstract was re-written including them. Please see lines 19-42 of the corrected document.

  • Methodology:
  1. Why was extraction carried out for only 35 minutes for pods??? 35 mins by conventional extraction is too short to exhaust the material for valuable metabolites.

Response: Extraction was carried out following the methodology of our previous research (Pedreschi et al., 2019) in which the kinetics of extraction were carried out to determine the equilibrium extraction time. In this sense, previous research performed by Chirinos et al., 2013, also found that 35 min at 60°C were proper extraction conditions for obtaining extracts with high antioxidant capacity and total polyphenol content.

  1. Mentioned at least references at first appearance for the methods of glucose, fructose, antioxidant activity and Total phenolic contents estimation. Authors should briefly describe tannin contents in Tara pods.

Response: References of antioxidant activity and Total phenolic contents estimation were included in the methodology section (Lines 166-176). Fructose and glucose were not evaluated in this research, because the focus of extract were antioxidant properties. Regarding to tannin contents and identification we proposed as part of future research to improve the chemical characterization of extracts considering their determination. Unfortunately, today is not possible to carry out experimental work because of the sanitary context.

  • Results
  1. Line 270-315, reference style is not as mentioned in guidelines to authors and the variations Line 332, ins??

Response: It was corrected (Lines 253-296).

  1. Table 3 caption is mentioned twice, moreover why table 3 does not indicate mean and SDs?? The manuscript lack in sufficient statistical analysis of the data, specifically tables of the manuscript.

Response: The caption of Table 3 was corrected. Means and CV are shown in table 3 (lines). The statistical analysis was deeply described in the methodology section (Lines 224-231).

In this sense, regarding to the mean and SD>, as we mentioned Table 3 presents the mean and the coefficient of variation (CV), where CV represents not only the relationship between the mean and the SD, but also the dispersion of the data regarding to the mean, that can be related with the experimental error.

Regarding to the statistical analysis, we carried out a factorial analysis which allows to evaluate the effect of the study variables on the different responses. This analysis can be evidenced in the significant differences found in table 3 and 4, which were represented by lowercase letters

  • Conclusions: The text is repetition of results. authors should mention concrete outcomes, future applicability and significance of the study. Discussion needs more evidence support from the literature.

Response: Conclusion was re-written considering all your comments and suggestions (Lines 402-420). The literature evidence support of discussion was completed (Lines 277-296).

Round 2

Reviewer 1 Report

The authors are slightly improved manuscript according to suggestions and the actual Covid 19 situation. The paragraphs at 282 and 294 lines are doubled. I recommend publication.

Best regards,

Author Response

The mistake was corrected.

Reviewer 3 Report

The authors have incorporated all the changes, advised by the reviewers.